# Trials of Commercial- and Wild-Type *Saccharomyces cerevisiae* Strains under Aerobic and Microaerophilic/Anaerobic Conditions: Ethanol Production and Must Fermentation from Grapes of Santorini (Greece) Native Varieties

Kalliopi Basa, Seraphim Papanikolaou *, Maria Dimopoulou, Antonia Terpou, Stamatina Kallithraka and George-John E. Nychas

Department of Food Science and Human Nutrition, Agricultural University of Athens, 75 Iera Odos, 11855 Athens, Greece; kbas1995@gmail.com (K.B.); mdimopoulou@uniwa.gr (M.D.); aterpou@agro.uoa.gr (A.T.); stamatina@aua.gr (S.K.); gjn@aua.gr (G.-J.E.N.)
* Correspondence: spapanik@aua.gr; Tel./Fax: +30-210-5294700

**Abstract:** In modern wine-making technology, there is an increasing concern in relation to the preservation of the biodiversity, and the employment of "new", "novel" and wild-type *Saccharomyces cerevisiae* strains as cell factories amenable for the production of wines that are not "homogenous", expressing their terroir and presenting interesting and "local" sensory characteristics. Under this approach, in the current study, several wild-type *Saccharomyces cerevisiae* yeast strains (LMBF Y-10, Y-25, Y-35 and Y-54), priorly isolated from wine and grape origin, selected from the private culture collection of the Agricultural University of Athens, were tested regarding their biochemical behavior on glucose-based (initial concentrations *ca* 100 and 200 g/L) shake-flask experiments. The wild yeast strains were compared with commercial yeast strains (*viz.* Symphony, Cross X and Passion Fruit) in the same conditions. All selected strains rapidly assimilated glucose from the medium converting it into ethanol in good rates, despite the imposed aerobic conditions. Concerning the wild strains, the best results were achieved for the strain LMBF Y-54 in which maximum ethanol production ($EtOH_{max}$) up to 68 g/L, with simultaneous ethanol yield on sugar consumed = 0.38 g/g were recorded. Other wild strains tested (LMBF Y-10, Y-25 and Y-35) achieved lower ethanol production (up to ≈47 g/L). Regarding the commercial strains, the highest ethanol concentration was achieved by *S. cerevisiae* Passion Fruit ($EtOH_{max}$ = 91.1 g/L, yield = 0.45 g/g). Subsequently, the "novel" strain that presented the best technological characteristics regards its sugar consumption and alcohol production properties (*viz.* LMBF Y-54) and the commercial strain that equally presented the best previously mentioned technological characteristics (*viz.* Passion Fruit) were further selected for the wine-making process. The selected must originated from red and white grapes (Assyrtiko and Mavrotragano, Santorini Island; Greece) and fermentation was performed under wine-making conditions showing high yields for both strains ($EtOH_{max}$ = 98–106 g/L, ethanol yield = 0.47–0.50 g/g), demonstrating the production efficiency under microaerophilic/anaerobic conditions. Molecular identification by rep-PCR carried out throughout fermentations verified that each inoculated yeast was the one that dominated during the whole bioprocess. The aromatic compounds of the produced wines were qualitatively analyzed at the end of the processes. The results highlight the optimum technological characteristics of the selected "new" wild strain (*S. cerevisiae* LMBF Y-54), verifying its suitability for wine production while posing great potential for future industrial applications.

**Keywords:** wild-type yeast strains; high-gravity fermentation; *Saccharomyces cerevisiae*; Assyrtiko; Mavrotragano; molecular identification; wine production

## 1. Introduction

The beginning of Greek viticulture dates back to the early Neolithic Age as favorable soil and climate of Greece have allowed the wide spread of wine-making [1]. The technologies of viticulture and wine-making were widely developed along the Mediterranean region since grapes are an excellent raw material for wine-making [2]. Nowadays, wine is an important component of the Mediterranean dietary tradition, while recent studies have moderated that its consumption reduces the incidences of coronary heart disease, atherosclerosis and protects against oxidative damage [3–5].

In the last decade, a significant number of studies targeted to improve the fermentation performance and productivity of wines and alcoholic beverages have been carried out [6–8]. An important factor in wine-making is the grape microbial ecosystem which is highly composed of diverse microorganisms, including yeasts, bacteria and fungi. The grape microbial ecosystem can result in a spontaneous wine fermentation which is a complicated process where many indigenous microorganisms may occur, including both *Saccharomyces cerevisiae* and non-*Saccharomyces* yeasts, lactic acid or acetic acid bacteria and fungi [9]. The final quality of wine is highly dependent on wine yeasts' metabolic activities; therefore, many wineries use selected yeast strains targeting to control fermentation and achieve high-quality wines [10].

Considering ethanol industrial production, yeasts represent the most significant microbial group providing high yields of productivity and also high tolerance and resistance to antagonistic microflora [11]. Biotechnological applications upon ethanol production mainly target bioprocess optimization technologies and application of selected/"optimized" strains. Regarding bioprocess optimization, research mainly focuses on the technological optimization of bioprocesses, such as the effect of the agitation/aeration upon the process, the "very high-gravity" application of the fermentation (*viz.* "the preparation and fermentation to completion of mashes containing 27 g or more dissolved solids per 100 g mash"), the utilization of "new" fermentation feedstocks (e.g., algae, crude glycerol, food-processing wastes, hemicellulose hydrolysates) and the development of innovative bioreactor designs [12–15]. On the other hand, biotechnological applications on bioethanol production relay on the yeast's "system biology" studies, including mostly screening and biochemical characterization of "novel" yeast strains and genetic engineering/random mutation/adaptive laboratory evolution strategies in order to "construct" "new" robust high-performing yeast strains with specified and desired fermentation characteristics [12,13,16], with a major topic currently developed being linked towards the isolation of novel robust high-performing microorganisms presenting technological interest [12,13,17]. Likewise, wine-making science is highly interested in the discovery of novel high-performing microorganisms aiming to be applied for wine-making providing quality wines with unique sensory characters [18,19].

It is widely known that the ordinary practice in wine-making facilities regarding the utilization of yeasts performing alcoholic fermentation refers to the employment of commercial "optimized" *Saccharomyces cerevisiae* strains, which results in the production of highly "homogenous" types of wines [20,21]. On the other hand, the use of "indigenous" ("local", *viz.* isolated from various "wine-type" products or various origins) strains amenable to be used as starters represents a potentially very useful tool to safeguard several types of sensory characteristics from specific regions, while it also demonstrates the potential of the biodiversity on the mentioned process [21–23]. Another approach demonstrates the use of *S. cerevisiae* yeast strains for aerobic fermentations targeting to obtain reduced ethanol yields in wines with acceptable volatile composition [24]. Moreover, in several cases, newly isolated (*viz.* non-commercial) *Saccharomyces cerevisiae* strains can present even better biotechnological properties compared to the commercial strains (i.e., higher vitality, killer factor, resistance to high concentrations of sugar, ethanol and $SO_2$, increased trehalose and glycogen cellular content), which could enhance their performance during the grape must fermentation process [21,22,25]. Therefore, according to the previously mentioned analysis, and in regard of the need in high-performing novel yeast strains,

while avoiding the long term and strain-stressful techniques of genetic engineering or the controversial techniques of mutation, the present study was conducted with the aim to assess the biochemical and technological properties of three commercial and four wild-type yeasts concerning comparison of strains' performance and productivity. Subsequently, two *S. cerevisiae* strains that were chosen based on their ethanol production capabilities (one commercial and one wild-type "novel" strain not previously systematically studied regarding its fermentation potential), were applied in a wine-making process, and applied for the fermentation of white and red musts from indigenous *Vitis vinifera* Greek grape varieties of the Santorini region. The strains were monitored via molecular analysis throughout fermentation targeting the evaluation of strains' adaptation, performance and productivity. Technological considerations regarding the performances of the yeast strains were critically considered and assessed.

## 2. Materials and Methods

### 2.1. Yeast Strains and Culture Conditions

Commercial- and wild-type yeast strains of the species *Saccharomyces cerevisiae* were selected and studied in synthetic media targeting application of selected yeast strains as starter cultures for wine-making of *Vitis vinifera* Greek grape varieties from the Santorini region. Specifically, the new isolated strains LMBF Y-10, Y-25, Y-35 and Y-54, originated from the culture collection of the Laboratory of Microbiology and Biotechnology of Foods (Department of Food Science and Human Nutrition, Agricultural University of Athens, Greece), originated from various types of food-stuffs (*viz.* commercial wine, grape musts and grapes) and not having been previously systematically studied regarding their fermentation potential, were used as cell factories in the present study. The commercial strains: Symphony, Cross X and Passion Fruit, were also used for comparison reasons. Strains were regenerated in YPDA slants (20 g/L glucose, 10 g/L yeast extract, 10 g/L peptone and 25 g/L agar) every 4 months to maintain the yeast viability [26]. Pre-cultures were performed in 250 mL non-baffled conical flasks filled with 50 mL of medium (YPD medium: 20 g/L glucose, 10 g/L yeast extract, 10 g/L peptone, pH $\approx$ 3.5) previously autoclaved at $T = 115\ ^\circ$C/1.5 atm for 15 min.

Aerobic experiments were performed in 250 mL (filled with 50 $\pm$ 1 mL medium) agitated flasks (use of a ZHWY-211B Rocking Incubator) at 180 $\pm$ 5 rpm in which commercial glucose (Hellenic Industry of Sugar SA, Orestiada, Greece) was used as carbon source. Glucose-based media presented the following salt composition in g/L: $KH_2PO_4$ 7.0; $Na_2HPO_4$ 2.5; $MgSO_4 \cdot 7H_2O$ 1.5; $CaCl_2 \cdot 2H_2O$ 0.15; $FeCl_3 \cdot 6H_2O$ 0.15; $ZnSO_4 \cdot 7H_2O$ 0.02; $MnSO_4 \cdot H_2O$ 0.06 [26]. The nitrogen sources used were peptone and yeast extract (at concentrations 3.0 and 3.0 g/L respectively), while initial glucose ($Glc_0$) was added to *ca* 100 g/L and *ca* 200 g/L. The pH value was adjusted to 3.5 $\pm$ 0.2, while incubation temperature $T = 30 \pm 1\ ^\circ$C was employed. Flasks were aseptically inoculated with 1 mL of yeast preculture (thus, a 2% *v/v* inoculation occurred). In the elaborated kinetics, each flask constituted the experimental point for the relevant experiments.

Microaerophilic/anaerobic trials were performed during wine-making experiments in static cultures [14]; grape must from two different *Vitis vinifera* Greek grape varieties of Santorini indigenous, Assyrtiko and Mavrotragano, were used for white and red wine production, respectively. The initial concentration of total reducing sugars (TS) for Assyrtiko must was 221.5 g/L, pH was =3.3 and total acidity was =6.0 g/L (expressed as tartaric acid) while for Mavrotragano must the respective values were 213.5 g/L, 3.4 and 5.8 g/L. In addition, SpringFerm™ (Fermentis, France) nutrients (0.2 g/L) were applied in each fermentation batch targeting enhanced fermentation rates [27]. Subsequently, alcoholic fermentations were carried out by applying the strains LMBF Y-54 and Passion Fruit under static cultures, ensuring microaerophilic (initially) and nearly anaerobic (after the initial steps of the fermentation and the $CO_2$ accumulation in the must and the bottle) conditions [12,14] in 1.0 L Duran bottles containing 500 mL of grape must. Alcoholic fermentations were performed at $T = 18\ ^\circ$C constant temperature incubating an initial

yeast population of $5 \times 10^6$ cfu per 500 mL of must (*viz.* 5 mL of exponential pre-culture in 500 mL of must). During fermentation, samples were obtained aseptically at 24-h intervals for further analysis, while sugar depletion signified the end of the wine-making process [7]. At the end of fermentation, each wine sample was treated with 0.08 g/L potassium metabisulfite (Fluka, Switzerland) and placed for 5 days in refrigerator storage ($T = 4\,°C$) targeting clarification. Finally, wine samples were transferred in sterile clean bottles and evaluated regarding their sensory attributes.

### 2.2. Analytical Methods

The whole content of the 250 mL flasks (*viz.* $50 \pm 1$ mL) or 5–10 mL of the content of static Duran bottles was collected at predetermined intervals to correctly assess the kinetic studies. Yeast biomass (for the shake-flask trials) was harvested by centrifugation at 9055 $g$, for 10 min at $T = 21 \pm 1\,°C$ (Suprafuge, Heraeus Sepatech), washed with distilled water and re-centrifuged again. Yeast cell concentration was determined gravimetrically by placement of wet biomass at $T = 95 \pm 5\,°C$ until constant weight (usually within $24 \pm 2$ h) and was expressed as dry cell weight (DCW) (X, g/L). Intra-cellular polysaccharides (IPS, expressed as % $w/w$ in DCW) were measured by collecting 0.05 g of dry biomass weighted in a precision scale of four decimal digits (Ker new 420-3NM) and placed in McCartney glass containers. The dried yeast cell mass was hydrolyzed using 10 mL of 2.5 M HCl at $T = 80\,°C$ for 30 min. The whole was neutralized to pH = 7.0 with 2.5 M NaOH and the volume was adjusted to 100 mL. Samples containing total sugars were then filtered (through No. 2 Whatman filters) and subjected to the DNS assay [28,29].

Ethanol, glucose, fructose and glycerol were quantified through high performance liquid chromatography (HPLC) analysis carried out in a Waters Association 600E apparatus equipped with a RI detector (Waters 410). A Rezex ROA-Organic Acid H$^+$ column (300 mm $\times$ 7.8 mm) (Phenomenex, Torrance, California, USA) was used for the separation of the compounds. The mobile phase was $H_2SO_4$ at 0.005 M. The column temperature was set at $T = 40\,°C$ with a flow rate of 0.5 mL/min. The injection volume was 20 μL [26]. For quantitative analysis, standard solutions of the compounds were prepared in pure water (Milli-Q, Merk) at various concentrations. The range of concentration used to build the calibration curve for each compound analyzed (*viz.* glucose, fructose, glycerol, ethanol) was between 0.0 and 20.0 g/L.

In the shake-flask experiments, dissolved oxygen concentration (DOC, in % $v/v$) was off-line determined using a selective electrode (OXI 96, B-SET, Germany) according to previously published procedure [29]. Before harvesting, the shaker was stopped and the probe was placed into the flask, after which the shaker was again switched on and the measurement was taken after DOC equilibration (within *ca* 10 min). In all experiments, and irrespective of the initial concentration of glucose set in the medium or the used strain, DOC values were for all culture phases $\geq 20\%$ $v/v$, indicating that in the shake-flask experiments performed, full aerobic conditions were maintained [26,29].

At the end of the microaerophilic/anaerobic trials, the volatiles of produced wines were determined using Gas Chromatography/Mass Spectrometry with Headspace Solid-Phase Micro-Extraction sampling (HS-SPME/GC–MS) [30]. Specifically, 2 mL of each wine sample was mixed with 7.5 mL of deionized water, 2 g of sodium chloride (dried at >100 °C prior to weighing), and 500 μL of 1,4-dioxane solution (1000 mg/L) as IS and placed in 20 mL glass vials. Each amber headspace vial was sealed with a cap (Teflon-lined septum) and placed in a water bath (40 °C) under constant stirring. Each vial was sealed with a silicone septum and placed in the water bath for 5 min reaching 40 °C under constant stirring (250 rpm). Subsequently, the SPME fibre (SPME; fibre DVB/CAR/PDMS, 2 cm; Sigma–Aldrich, Germany) was exposed to the headspace for 30 min at 40 °C under constant stirring. Then the fibre containing the absorbed volatiles was exposed in the injection port of the chromatograph (GC), in a split mode (split ratio 1/10), at 240 °C for 5 min. The chromatograph (GCMS-QP2010 Ultra, Shimadzu Inc., Kyoto, Japan) was equipped with a DB-Wax capillary column (30 m, 0.25 μm film thickness, Agilent, Santa Clara, California,

USA) and was applied as carrier gas. The mass spectrometer operated in an electron ionization mode, at an ionization energy of 70 eV and 4 at 0–300 m/z mass scan range. The source and interface temperatures were set at 200 °C and 240 °C, respectively.

For the identification of volatiles, the following were compared: (i) retention index (RI) based on the homologous series of C8-C24 n-alkanes with those of available authentic compounds and those available in the NIST14 library (NIST, Gaithersburg, Maryland, USA), and (ii) AMDIS software (v. 2.65 build 116.66) was employed, based on retention time and mass spectra, with a parallel use of NIST library as confirmation. The analysis of volatile profile of wines was based on the absolute values of the peak area of each compound, and they were expressed as a percentage of the total peak area [(compound peak area/sum of peak areas) × 100].

### 2.3. Molecular Identification of Yeast Cells

For the non-aseptic trials performed in the Duran bottles, samples were analyzed in the beginning, the middle and in the end of the fermentation process, for yeast identification at strain level. One milliliter of each sample was aseptically transferred in 9 mL of sterile $\frac{1}{4}$ Ringer's solution and serially diluted in the same diluent. Each dilution with 0.1 mL was spread at yeast extract peptone dextrose (YPD) agar plates and inoculated at $T$ = 28 °C for 48 h. A percentage of 20% of colonies was picked from the appropriate dilution and transferred at 10 mL of YPD broth according to the representative sampling scheme of Harrigan and McCance, which is still applied over time [31,32]. Total DNA was extracted from each culture [33]. Briefly, one milliliter of overnight culture was centrifuged (14,000 rpm) for 5 min at 4 °C then the pellet was resuspended in 0.5 mL buffer solution (1 M sorbitol, 0.1 M EDTA, pH 7.5) containing lyticase (2.5 U/mL) (lyticase from *Arthrobacter luteus*, Sigma–Aldrich, Germany) for yeast cell lysis. After centrifugation, the pellet was resuspended in 0.5 mL of buffer (50 mM Tris–HCl, 20 mM EDTA, pH 7.4) and incubated for 30 min at 65 °C with 50 μL of 10% SDS solution. Then, each sample was mixed with 0.2 mL potassium acetate (5M) (Merck) for 30 min and centrifuged (14,000 rpm) for 10 min at 4 °C. The supernatant was precipitated with 1 mL ice-cold isopropanol (Applichem) and then centrifuged (14,000 rpm) for 10 min at 4 °C. The final pellet was dried and resuspended in 50 μL sterile ddH20. The quantification and quality control of DNA extract was performed by means of a nanophotometer (Implen, Germany) at wavelengths of 260, 280 and 230 nm. Pure cultures of the inoculated yeast strains, *S. cerevisiae* LMBF Y-54 and *S. cerevisiae* Passion Fruit were used as controls strains for the molecular characterization.

A rep-PCR method was applied in order to unambiguously discriminate genotypes of different species and reach strain level. PCR-fingerprinting was performed with the primers $(GTG)_5$ [34]. The reaction involved initial denaturation at 94 °C for 4 min, followed by 35 cycles of the series of 94 °C for 15 s, 55 °C for 45 s, and 72 °C for 90 s, with a final cycle at 72 °C for 15 min. Amplification was carried out in a thermocycler (Applied Biosystems, Bedford, MA, USA). PCR products were separated by electrophoresis in a 1.3% agarose gel, 1 × TAE (40 mM Tris-acetate, 1 mM EDTA, pH 8.2) buffer at 80 V for 120 min, then were stained with ethidium bromide solution (1%) and finally digitalized under UV light (GelDoc system, Bio-Rad, Hercules, CA, USA). The similarity among digitalized profiles was calculated using the Pearson correlation and an average linkage (UPGMA) dendrogram was derived from the profiles. The 1 Kb (Invitrogen, Waltham, Massachusetts, USA) molecular weight marker was used to compare the sizes of the bands. Cluster analysis was performed using the Bionumerics software version 6.1 (Applied Maths, Sint-Martens-Latem, Belgium).

### 2.4. Sensory Assessment

Samples were evaluated by a group of 12 trained panelists with previous experience in wine sensory analysis [35–37]. The tests were conducted from 11:00 a.m. to 13:00 a.m. in individual booths. In more detail, each sample was served in a completely randomized presentation order and was evaluated in triplicate by each panelist. The judges were

provided with 30 mL of each sample in standard wine glasses, marked with three-digit random numbers, at room temperature ($T$ = 18–20 °C). The following olfactory attributes were evaluated: floral, fresh fruits, dry fruits, reduction, odor of oxidation, aroma intensity and overall aroma quality using a 5-point scale. Zero intensity of the attributes was marked on the left end of the scale whereas maximum intensity was marked on the right.

### 2.5. Data Analysis

Each experimental point of all the kinetics presented in the tables and figures is the mean value of two independent determinations, while the standard error (SE) for most experimental points was ≤17%. Data were plotted using Kaleidagraph 4.0 Version 2005 showing the mean values with the standard error mean.

Regarding statistical analysis of volatile contents and sensory results, analysis of variance (ANOVA) was performed using Statistica V.7 (Statsoft Inc., Tulsa, OK, USA) to determine whether the mean values differed between samples. Tukey's HSD was used as comparison tests when samples were significantly different after ANOVA ($p < 0.05$).

### 2.6. Nomenclature

X: Dry cell weight (DCW; total microbial mass) (g/L); Glc: Glucose (g/L); Fru: Fructose (g/L); TS: Total sugars (g/L); EtOH: ethanol (g/L); Glyc: Glycerol (g/L); IPS: intra-cellular polysaccharides (g/L); t: fermentation time (h); $Y_{EtOH/Glc}$: yield of ethanol production with respect to glucose consumed (g/g); $Y_{EtOH/TS}$: yield of ethanol production with respect to total sugars consumed (g/g); $Y_{IPS/X}$: intra-cellular polysaccharides in DCW (%, $w/w$); indices 0, max and cons show the initial and maximum and the consumed quantity of the elements in the experiments carried out; DOC: dissolved oxygen concentration (%, $v/v$).

## 3. Results and Discussion

### 3.1. Commercial- and Wild-Type Yeast Strains Perfrormemces in Synthetic Medium/Aerobic Experiments and Yeast Selection

Ethanol production in wine is based on the ability of yeast strains to catabolize six-carbon molecules present in must into ethanol, a two-carbon compound [11]. The bio-processes where yeasts convert glucose to ethanol are known as the glycolytic pathway followed by ethanol fermentation [38]. Fermentation (*viz.* "anaerobic"-type transformation of glucose into ethanol) may occur despite the presence of $O_2$ in the culture medium in significant concentrations when the initial concentration of the employed carbohydrate (i.e., glucose and/or fructose and/or sucrose) is higher than a "critical" value [39]. In fact, for a remarkable number of yeast species (the so-called "conventional" yeasts), even with the significant presence of oxygen in the fermentation medium (i.e., DOC values ≥ 20% $v/v$ and in some cases >50% $v/v$), if sugar concentration is higher than a critical (and in many instances not very high) concentration (e.g., *ca* 9 g/L or even lower), respiration is impossible; furthermore, despite the oxygen saturation conditions imposed, the microorganism shifts its metabolism completely towards the fermentative pathway and the subsequent biosynthesis and accumulation of ethanol into the medium. This phenomenon is known as the "Crabtree effect" (named after the English biochemist Herbert Grace Crabtree), Pasteur contrary effect, or as catabolic repression by glucose [12,40]. Specifically, for the mentioned type of yeasts (the "Crabtree"-positive ones), in somehow elevated sugar concentrations imposed in the culture medium, the mitochondria degenerate, the proportion of cellular sterols and fatty acids decrease and both the enzymes involved in the oxidative part of the metabolism (namely the Krebs cycle and the oxidative phosphorylation chain) and the constituents of respiratory chains are subjected to catabolite repression, leading to the elaboration of the ethanol fermentation despite oxygen-sufficient culture conditions [12,40,41].

All yeast strains tested in shake-flask trials converted glucose into ethanol (EtOH) and dry yeast biomass (X), despite aerobic conditions imposed into the medium (in all trials and irrespective of the strain, the initial glucose concentration and the culture time, the DOC was always ≥20% $v/v$, indicating sufficient aerobic conditions into the shake-flask

environment [7,29,41], and the obtained results are depicted in Table 1(a) ($Glc_0 \approx 100$ g/L) and Table 1(b) ($Glc_0 \approx 200$ g/L). In all trials performed, non-negligible quantities of glucose were relatively rapidly consumed and converted into DCW and (mostly) ethanol; therefore, all selected yeast strains were referenced as positive to "Crabtree effect" in accordance with the literature [12], that by far considers the *S. cerevisiae* species as the most "classic" yeast strain in which this phenomenon occurs [12,40–44]. It is interesting to indicate that in the trials with $Glc_0 \approx 100$ g/L, with the exception of the wild-type non-commercial LMBF Y-54 strain that consumed the majority of glucose quantity that was found in the growth medium, all other LMBF Y- strains did not consume all available sugar quantity of the medium, whereas further incubation did not lead to increased glucose assimilation, but to degradation (oxidation) of ethanol, that in most cases was not accompanied by a DCW concentration increase (thus no diauxic growth occurred). Concerning X production of the wild-type non-commercial yeasts growing on glucose at $Glc_0 \approx 100$ g/L, by far the highest biomass producer was the strain LMBF Y-54 (X = 7.1 g/L; see Table 1). The $X_{max}$ value of this strain was recorded at t = 121 h, being =10.5 g/L (kinetics non shown; this value was obtained after ethanol oxidation that occurred when glucose had been depleted from the medium). In fact, it is the so-called "ethanol make–accumulate–consume" phenomenon. This phenomenon (in fact a metabolic adaptation "strategy" developed by *Saccharomyces* strains under aerobic conditions) relies on the evolution of *Saccharomyces* cultures against their competitors, as ethanol is toxic to most other microbes. Therefore, it is considered in a (non-aseptic) sugar- and oxygen-rich environment that *Saccharomyces* strains eliminate their competitors by producing ethanol, but in a next fermentation step, they consume the previously generated ethanol, in order to create further DCW or maintain the already existing one (consumption of ethanol for energy of maintenance requirements) [12,41]. Alcohol dehydrogenase (Adh) catalyzes the acetaldehyde-to-ethanol conversion in both directions. Genes ADH1 (expressed constitutively) and ADH2 (expressed only when the internal sugar concentration drops) encode cytoplasmic Adh activity [12,40,41,43]. Concerning the commercial strains: Passion Fruit, Symphony and Cross X; all of these microorganisms converted glucose rapidly into ethanol with the highest quantities of glucose (*ca* 90% *w/w* of the available sugar) having been assimilated within the first 24–36 h after inoculation (Table 1(a)). In the trials with $Glc_0$ adjusted to around 100 g/L, the conversion yield $Y_{EtOH/Glc}$ presented variable values, with some wild-type strains (i.e., LMBF Y-25 and LMBF Y-35) presenting excellent values ($Y_{EtOH/Glc} \geq 0.45$ g/g, *viz.* $\geq 88\%$ of the maximum theoretical yield that is =0.51 g/g [12,13,41,43]). On the other hand, in similar types of experiments performed with other wild-type ("novel") or commercial-type *Saccharomyces cerevisiae* strains cultured in shake-flask trials on previously pasteurized natural grape musts under aerobic conditions, the conversion achieved yields $Y_{EtOH/TS}$ ranged between 0.28 and 0.40 g/g [24], that were values slightly or somehow lower compared to those obtained in the present study (Table 1). This indicates the potential of the employed strains in the current investigation regarding the conversion of glucose into ethanol under aerobic conditions. The Crabtree-effect (*viz.* the production of ethanol from glucose or other sugar fermentation under full aerobic conditions) has been studied in other conventional but non-*Saccharomyces* yeast strains, and the conversion yield $Y_{EtOH/TS}$ has been revealed to be 0.10 g/g for *Candida diddensiae*, 0.27 g/g for *Candida tropicalis* and 0.28 g/g for *Candida zemplinina* [44,45].

**Table 1.** Experimental results originated from kinetics of *S. cerevisiae* wild strains (LMBF Y-10, Y-25, Y-35 Y-54) and commercial strains (Symphony, Passion Fruit, Cross X) growing in shake-flask glucose-based synthetic media at initial glucose concentration $\approx$ 100 g/L (a) and $\approx$200 g/L (b) when the maximum concentration of ethanol (EtOH$_{max}$) was achieved. Representations of dry biomass (X, g/L), glucose consumed (Glc$_{cons}$, g/L), ethanol produced (EtOH, g/L) and ethanol conversion yield per sugar consumed ($Y_{EtOH/Glc}$, g/g). Each experimental point is the mean value of 2 independent measurements (SE for most experimental points $\leq$ 17%).

**(a).**

| Strain | Time (h) | Glc$_{cons}$ (g/L) | X (g/L) | EtOH (g/L) | Glyc (g/L) | $Y_{EtOH/Glc}$ (g/g) |
|---|---|---|---|---|---|---|
| *S. cerevisiae* LMBF Y-35 | 72.0 | 66.9 | 2.5 | 30.1 | 1.4 | 0.45 |
| *S. cerevisiae* LMBF Y-25 | 51.0 | 57.0 | 2.9 | 26.7 | 2.7 | 0.47 |
| *S. cerevisiae* LMBF Y-54 | 52.0 | 112.2 | 7.1 | 41.0 | 1.0 | 0.37 |
| *S. cerevisiae* LMBF Y-10 | 48.0 | 78.0 | 3.2 | 30.4 | 0.9 | 0.39 |
| *S. cerevisiae* Symphony | 36.0 | 93.0 | 5.9 | 32.6 | 0.5 | 0.35 |
| *S. cerevisiae* Cross X | 34.0 | 89.1 | 4.0 | 40.0 | 1.9 | 0.45 |
| *S. cerevisiae* Passion Fruit | 24.0 | 92.9 | 5.1 | 43.1 | 1.6 | 0.46 |

**(b).**

| Strain | Time (h) | Glc$_{cons}$ (g/L) | X (g/L) | EtOH (g/L) | Glyc (g/L) | $Y_{EtOH/Glc}$ (g/g) |
|---|---|---|---|---|---|---|
| *S. cerevisiae* LMBF Y-35 | 95.0 | 166.8 | 2.7 | 40.9 | 6.4 | 0.25 |
| *S. cerevisiae* LMBF Y-25 | 68.0 | 165.5 | 2.6 | 47.4 | 5.2 | 0.29 |
| *S. cerevisiae* LMBF Y-54 | 168.0 | 179.3 | 6.1 | 68.0 | 6.3 | 0.38 |
| *S. cerevisiae* LMBF Y-10 | 72.0 | 117.1 | 3.9 | 40.0 | 3.1 | 0.34 |
| *S. cerevisiae* Symphony | 48.0 | 214.6 | 8.1 | 60.9 | 1.4 | 0.28 |
| *S. cerevisiae* Cross X | 72.0 | 178.6 | 6.3 | 82.2 | 4.4 | 0.46 |
| *S. cerevisiae* Passion Fruit | 76.0 | 200.5 | 6.4 | 91.1 | 3.8 | 0.45 |

Adaptation to higher Glc$_0$ concentrations imposed in the medium (*viz. ca* 200 g/L) resulted in significant quantities of assimilated glucose irrespective of the strains (commercial or wild-type ones) implicated as microbial cell factories of the process (Table 1(b)). Apart from the strain LMBF Y-10, in all other cases, consumed Glc quantities $\geq$ 160 g/L were recorded, suggesting the potential of the employed strains towards the so-called "very-high-gravity" alcoholic fermentation process [12,13,26]. It is also interesting to indicate that despite the significant consumption of glucose that was reported for many of the trials, the conversion yield $Y_{EtOH/Glc}$ was in a number of cases lower than the respective one reported for the experiments with Glc$_0$ $\approx$ 100 g/L (see Table 1). This has also been reported in other cases in which alcoholic fermentations had been performed in media in which all other culture parameters and components (i.e., initial nitrogen) remained constant and only glucose concentration increased [42,43], exactly as in the current investigation, demonstrating that potentially the conversion yield $Y_{EtOH/Glc}$ is negatively correlated with the increase in carbon excess in the medium (increment of Glc$_0$ concentration in the medium with the initial nitrogen remaining constant, evidently increases the initial C/N and, thus, the carbon excess in the medium [28,29,42]). On the other hand, given the fact that at the increasing Glc$_0$ concentrations, the initial concentration of glucose imposed on the medium is rather high ($\approx$200 g/L), besides potential application of nitrogen-limited conditions, other parameters such as oxidative stress and osmotic stress, that are both linked with high Glc$_0$ concentration media [46], may have negative impact upon the decrease in the $Y_{EtOH/Glc}$ values with a glucose concentration increment in the medium. In fact, as far as the ethanol fermentation under high Glc$_0$ concentrations by *Saccharomyces cerevisiae* strains is concerned, to alleviate the various environmental stresses imposed by the increased initial

concentrations of sugar, trials in media rich in $Mg^{2+}$ and organic nitrogen (i.e., peptone) seem of importance in order to achieve high ethanol concentrations and yields [47].

In all cases and in all fermentations performed, the intra-cellular polysaccharides were quantified, and $Y_{IPS/X}$ values ranging between 4.2–11.0% *w/w* were recorded for all strains, all fermentation periods and all culture conditions. The quantity of IPS per DCW (%, *w/w*) did not increase as the $Glc_0$ concentration in the medium increased, suggesting that under the present culture conditions and for the given strains tested, there was not any shift towards the synthesis of endopolysaccharides due to the increase in carbon excess in the medium [29,42]. This result does not comply with previously published data; in fact, in other studies in which yeasts showed both Crabtree-negative (i.e., *Rhodosporidium toruloides, Yarrowia lipolytica, Cryptococcus curvatus*) and Crabtree-positive (i.e., *Saccharomyces cerevisiae*) effect, the utilization of relatively high initial concentrations of sugar (i.e., ≥20 g/L) and the employment of carbon excess conditions (*viz.* the utilization of media in which the initial ratio of C/N in somehow high, i.e., ≥40–50 moles/moles) seem necessary in order to increase the quantity of IPS per unit of DCW [29,48–50], with $Y_{IPS/X}$ values in some cases being ≥55% *w/w*. It is finally noted that similar types of experiments performed by a wide range of wild-type and commercial *Saccharomyces cerevisiae* strains cultured under aerobic conditions on YPD medium resulted in the synthesis of intra-cellular polysaccharides (mostly trehalose and glycogen) with $Y_{IPS/X}$ values of 9.0–18.0% *w/w* (values somehow similar with those reported in the current investigation) [25].

### 3.2. Wine-Making of Selected Strains/Microaerophilic/Anaerobic Experiments

As mentioned above, "novel" (*viz.* newly isolated and, therefore, non-commercial) *Saccharomyces cerevisiae* strains can present in some instances equal (or even better) biotechnological properties compared to the commercial strains (i.e., higher vitality, resistance to high concentrations of sugar, high potential of sugar uptake, high ethanol production, equal of increased content of intra-cellular polysaccharides), while these properties, linked also to the potential for the production of wines with not "homogenous" organoleptic characters, can present significant importance for the wine-making industries. One of the "novel" strains of the current study (*viz.* the strain LMBF Y54) presented very interesting technological characteristics regarding its sugar consumption and alcohol production properties. In fact, this wild-type strain was the best amongst the "novel" studied strains regarding its potential upon the previously mentioned characteristics (see Table 1). For this reason, this wild-type strain was chosen for further trials. Equally, one of the commercial strains that presented the best previously mentioned technological characteristics (*viz.* the strain Passion Fruit) was also selected (see Table 1). Both strains were further studied in wine-making conditions. These strains were cultivated in static conditions that were performed under microaerophilic (initially) and self-generated anaerobic (after the initial steps of the fermentation and the subsequent $CO_2$ production in the must and the flask) conditions (see also [14]) in 1.0 L Duran bottles in wine-making conditions using grape musts of the varieties Assyrtiko and Mavrotragano, and the achieved results are depicted in Table 2.

The wild yeast strain, *Saccharomyces cerevisiae* LMBF Y-54, achieved maximum sugar consumption in Mavrotragano must after *ca* 13 days of fermentation (Table 2). Kinetic analysis for both fermented musts demonstrated that glucose and fructose were assimilated with almost equivalent assimilation rates (see example in Assyrtiko fermentation in Figure 1a), that is a not frequently a phenomenon seen in similar types of fermentations [51].

**Table 2.** Experimental results originated from kinetics of *S. cerevisiae* LMBF Y-54 and Passion Fruit growing on Assyrtiko and Mavrotragano must at points when a maximum concentration of ethanol (EtOH$_{max}$) was achieved. Representations of residual glucose (Glc, g/L), residual fructose (Fru, g/L), total sugars consumed (TS$_{cons}$, g/L), ethanol produced (EtOH, g/L) and ethanol conversion yield per total sugars consumed ($Y_{EtOH/Glc}$, g/g). Each experimental point is the mean value of 2 independent measurements (SE for most experimental points $\leq$ 17%).

| | Time (h) | Glc (g/L) | Fru (g/L) | TS$_{cons}$ (g/L) | EtOH (g/L) | Glyc (g/L) | $Y_{EtOH/TS}$ (g/g) |
|---|---|---|---|---|---|---|---|
| Assyrtiko must | | | | | | | |
| Passion Fruit; TS$_0 \approx$ 221.5 g/L | 310 | 3.7 | 8.0 | 209.8 | 102.7 | 2.2 | 0.49 |
| LMBF Y-54; TS$_0 \approx$ 221.5 g/L | 310 | 3.9 | 3.7 | 213.9 | 106.3 | 3.5 | 0.50 |
| Mavrotragano must | | | | | | | |
| Passion Fruit; TS$_0 \approx$ 213.5 g/L | 286 | 1.8 | 5.8 | 205.9 | 97.8 | 5.2 | 0.47 |
| LMBF Y-54; TS$_0 \approx$ 213.5 g/L | 240 | 2.2 | 1.9 | 209.4 | 99.7 | 5.9 | 0.48 |

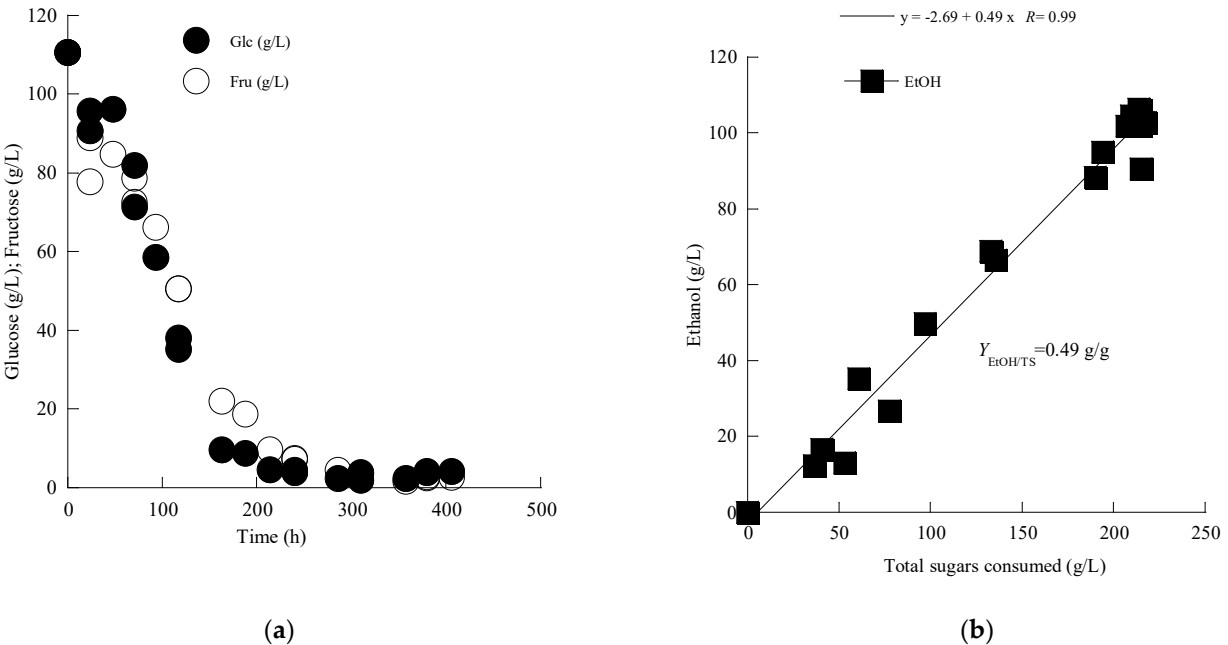

(**a**)     (**b**)

**Figure 1.** (**a**) Kinetics of glucose (Glu) and fructose (Fru) of the *S. cerevisiae* LMBF Y-54 strain in must from the grape variety Assyrtiko. Each experimental point is the mean value of 2 independent measurements (SE for most experimental points $\leq$ 17%). (**b**) Representation of ethanol produced vs. total sugars consumed of *S. cerevisiae* strain LMBF Y-54 in must from the grape variety Assyrtiko. for the whole set of data. Each experimental point is the mean value of 2 independent measurements (SE for most experimental points $\leq$ 17%).

Moreover, Mavrotragano fermentation was accompanied by the slightly lower EtOH$_{max}$ concentration achieved compared to Assyrtiko grape must (Table 2). This outcome could be attributed to the initial higher sugar content of Assyrtiko grape must. The global conversion yield of ethanol produced per unit of total sugar consumed ($Y_{EtOH/TS}$, in g/g), calculated by linear regression of the concentration of EtOH produced as a function of the quantity of TS consumed for the whole set of experimental data (see Figure 1b, for Assyrtiko fermentation) demonstrates very similar values with those reported in Table 2, that were calculated on the basis of the experimental point showing the highest ethanol concentration achieved and the respective remaining total sugar concentration value (*viz.* $Y_{EtOH/TS} = \frac{EtOH_{max}}{TS_0 - TS_t}$, where TS$_t$

is the remaining total sugar concentration in the must when the EtOH$_{max}$ concentration is achieved; see Table 2).

A well-established sugar consumption capability is a crucial step for wine fermentation as high sugar contents can cause sluggish or stuck fermentation rates. On the other hand, the final sugar content can cause undesirable sweetness in wines or even unwanted fermentation during storage. As a result, low final sugar content and high ethanol yield are mostly wanted in wine-making [7]. From all the above-mentioned analysis and taking into consideration the results achieved with the wild-type LMBF Y-54 strain (see Table 2), the high potential of this strain towards the production of wines with desired characteristics was demonstrated. Regarding glycerol production, a gradual increase was noted achieving a maximum value of 5.9–6.1 g/L by the end of fermentation for the case of Mavrotragano must, whereas for the Assyrtiko must, the respective concentrations at the end of growth were 3.5–3.8 g/L.

The commercial yeast strain, *Saccharomyces cerevisiae* Passion Fruit, presented similar experimental results with the wild-type LMBF Y-54 strain. In both types of grape musts, fermentations were carried out (see Table 2). On the other hand, in disagreement with the results for the LMBF Y-54 strain, the commercial strain demonstrated a higher assimilation rate of glucose compared to fructose (see example in Assyrtiko fermentation in Figure 2a), in accordance with the results reported for several wild-type or commercial yeast strains in wine-making conditions [51–53]. As previously mentioned, the global conversion yields $Y_{EtOH/TS}$ (see the yield for the case of Assyrtiko fermentation represented in Figure 2b) were almost the same as those that were calculated based on the experimental point showing the highest ethanol concentration achieved (see as previously: $Y_{EtOH/TS} = \frac{EtOH_{max}}{TS_0 - TS_t}$; Table 2). Finally, glycerol production showed an upward trend, with a maximum concentration of 5.2–5.5 g/L after 406 h of fermentation for the case of Mavrotragano must, whereas for the Assyrtiko must, the respective values at the end of growth were 2.2–2.6 g/L.

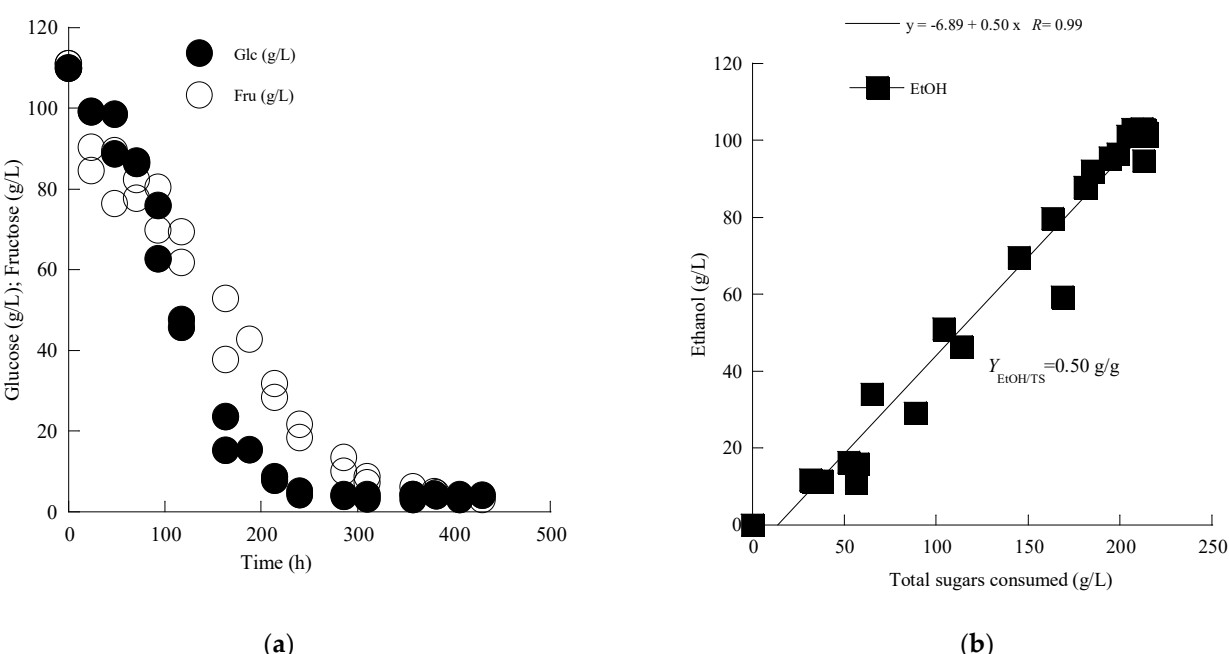

(a)            (b)

**Figure 2.** (**a**) Kinetics of glucose (Glu) and fructose (Fru) of the *S. cerevisiae* Passion Fruit strain in must from the grape variety Assyrtiko. Each experimental point is the mean value of 2 independent measurements (SE for most experimental points ≤ 17%). (**b**) Representation of ethanol produced vs. total sugars consumed of the *S. cerevisiae* Passion Fruit strain in must from the grape variety Assyrtiko. for the whole set of data. Each experimental point is the mean value of 2 independent measurements (SE for most experimental points ≤ 17%).

According to molecular typing with rep-PCR, all isolates showed the same finger-printing with the *Saccharomyces cerevisiae* inoculated strain in Assyrtiko must. The cluster analysis of the data set allowed the recognition of two main groups, branching at a similarity value of 90.0% in the case of the LMBF Y-54 strain and 87.9% in the case of the Passion Fruit strain (Figures 3 and 4). Likewise, PCR fingerprinting of *Saccharomyces cerevisiae* isolates with (GTG)5 in Mavrotragano must fermentation indeed yielded low genetic polymorphism for both inoculated strains. The obtained molecular patterns for the 30 isolates created two main groups, sharing a similarity value of 96.8% when LMBF Y-54 was inoculated and 95.8% when Passion Fruit was inoculated (Figures 5 and 6).

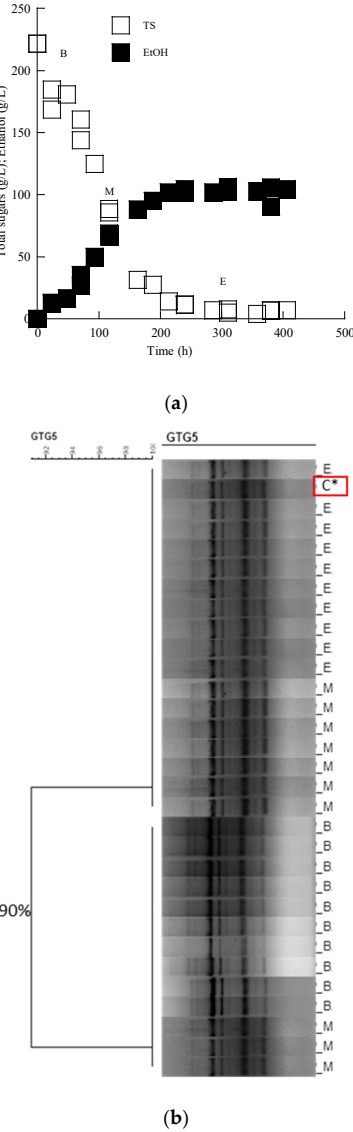

**(a)**

**(b)**

**Figure 3.** (**a**) Sugar and ethanol evolution of the *S. cerevisiae* LMBF Y-54 strain in must from the grape variety Assyrtiko. Each experimental point is the mean value of 2 independent measurements (SE for most experimental points ≤ 17%). (**b**) Dendrogram generated after cluster analysis of the digitized (GTG)5-PCR fingerprints of the 31 isolates from the beginning (B), middle (M) and end (E) of the fermentation kinetics in the grape must media. The control condition (C*) is the relative rep-PCR profile of the pure culture of the *S. cerevisiae* LMBF Y-54 strain. The dendrogram was constructed using the unweighted pair-group method using arithmetic averages with correlation levels expressed as percentage values of the Pearson correlation coefficient.

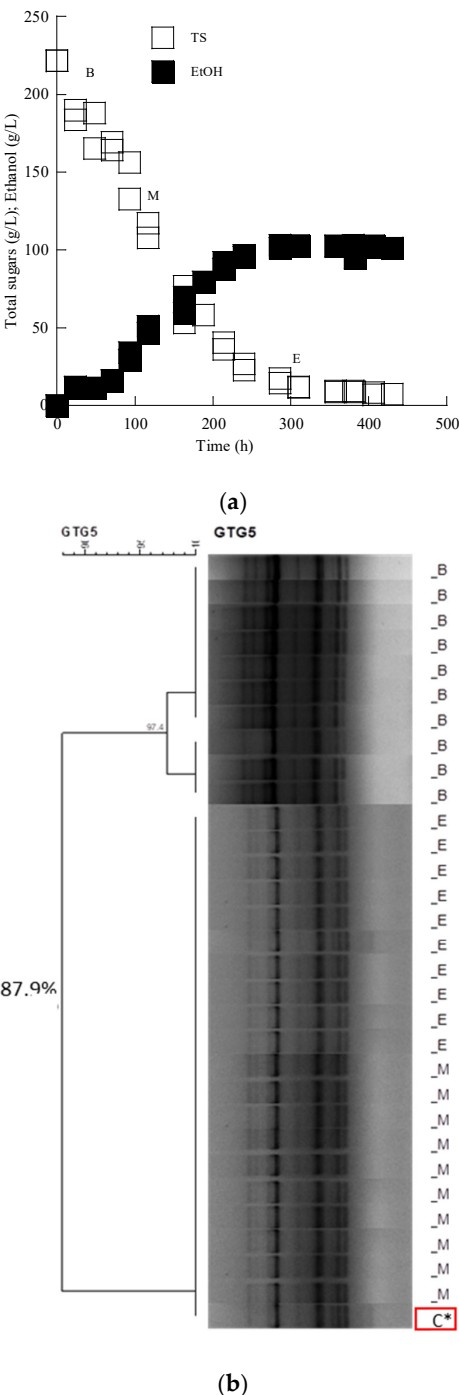

(**a**)

(**b**)

**Figure 4.** (**a**) Sugar and ethanol evolution of the *S. cerevisiae* Passion Fruit strain in must from the grape variety Assyrtiko. Each experimental point is the mean value of 2 independent measurements (SE for most experimental points ≤ 17%). (**b**) Dendrogram generated after cluster analysis of the digitized (GTG)5-PCR fingerprints of the 31 isolates from the beginning (B), middle (M) and end (E) of the fermentation kinetics in the grape must media. The control condition (C*) is the relative rep-PCR profile of the pure culture of the *S. cerevisiae* Passion Fruit strain. The dendrogram was constructed using the unweighted pair-group method using arithmetic averages with correlation levels expressed as percentage values of the Pearson correlation coefficient.

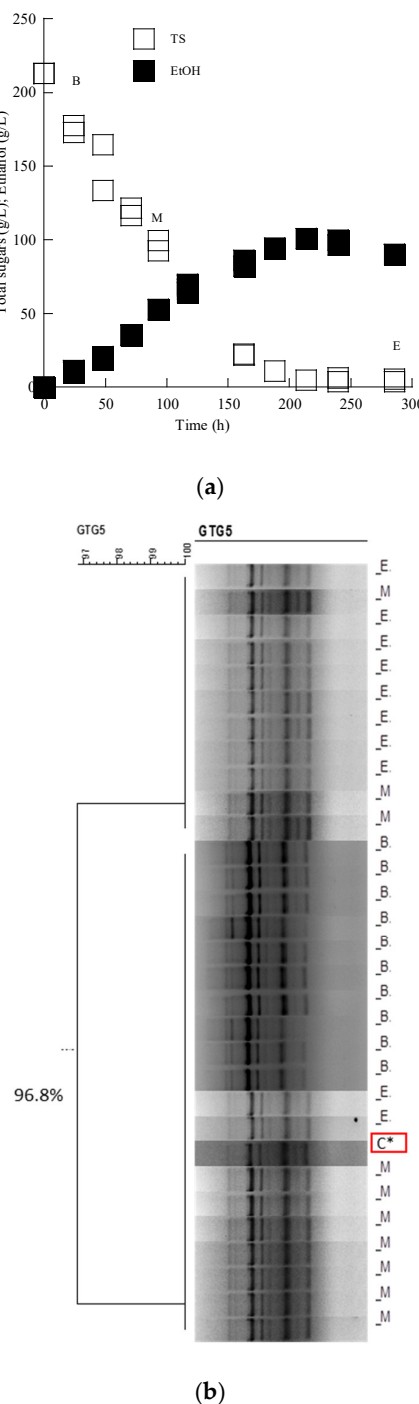

(**a**)

(**b**)

**Figure 5.** (**a**) Sugar and ethanol evolution of the *S. cerevisiae* LMBF Y-54 strain in must from the grape variety Mavrotragano. Each experimental point is the mean value of 2 independent measurements (SE for most experimental points ≤ 17%). (**b**) Dendrogram generated after cluster analysis of the digitized (GTG)5-PCR fingerprints of the 31 isolates from the beginning (B), middle (M) and end (E) of the fermentation kinetics in the grape must media. The control condition (C*) is the relative rep-PCR profile of the pure culture of the *S. cerevisiae* LMBF Y-54 strain. The dendrogram was constructed using the unweighted pair-group method using arithmetic averages with correlation levels expressed as percentage values of the Pearson correlation coefficient.

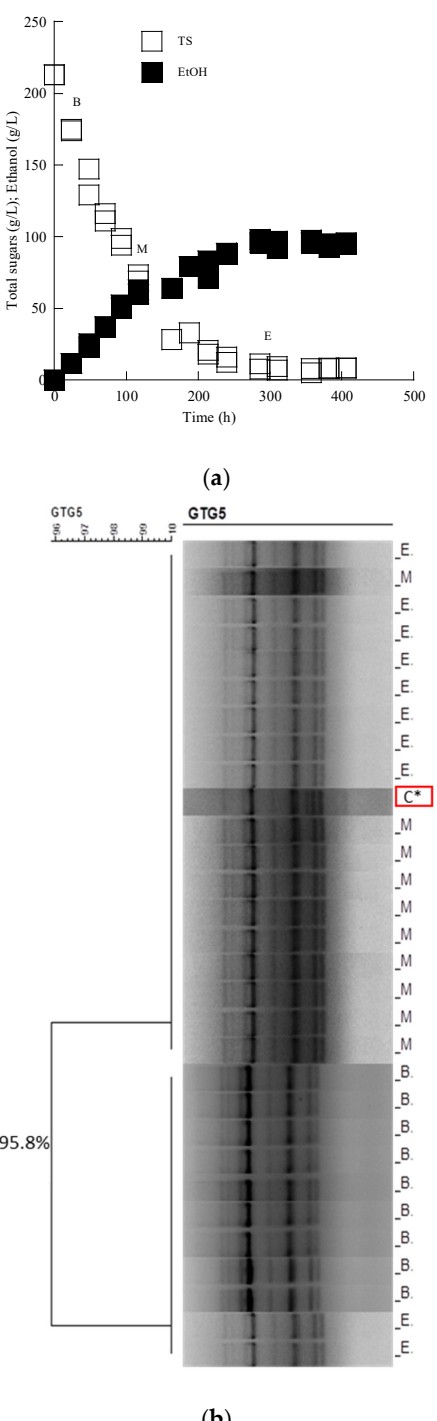

(**a**)

(**b**)

**Figure 6.** (**a**) Sugar and ethanol evolution of *S. cerevisiae* Passion Fruit in must from the grape variety Mavrotragano. Each experimental point is the mean value of 2 independent measurements (SE for most experimental points ≤ 17%). (**b**) Dendrogram generated after cluster analysis of the digitized (GTG)5-PCR fingerprints of the 31 isolates from the beginning (B), middle (M) and end (E) of the fermentation kinetics in the grape must media. The control condition (C*) is the relative rep-PCR profile of the pure culture of the *S. cerevisiae* Passion Fruit strain. The dendrogram was constructed using the unweighted pair-group method using arithmetic averages with correlation levels expressed as percentage values of the Pearson correlation coefficient.

The implicated strains (commercial and wild-type) presented significant ethanol production during growth of sugar-based media under aerobic and microaerophilic/anaerobic

experiments. It appears that the trials under microaerophilic/anaerobic conditions favored the production of ethanol, with this type of result being strain dependent according to the results found in the literature [7,24]. For instance, Tronchoni et al. [24], in accordance with our results, demonstrated that the cultures of various *Saccharomyces cerevisiae* strains performed on previously pasteurized grape musts under aerobic conditions was always accompanied by the biosynthesis of ethanol with lower $Y_{EtOH/TS}$ values, compared to equivalent anaerobic experiments [24]. In fact, according to the authors, this was a strategy developed in order to produce wines under aerobic conditions, since this method was proposed to reduce the ethanol content of the produced wine. It is noted that low alcohol-titer wines ($\leq$10.5% *v/v*) are novel products that due to multiple reasons (modern lifestyle, social reasons, economic motives, etc.) have been gradually gaining the interest of consumers and market in the last decade [24]. On the other hand, in other cases, cultures of wild-type *Saccharomyces cerevisiae* strains on grape musts or glucose-enriched media under aerobic conditions resulted in the production of ethanol in substantial quantities (in some cases in concentrations $\geq$ 110 g/L) with simultaneous very high $Y_{EtOH/TS}$ values (i.e., $\geq$0.40 g/g, in some instances these values were $\approx$0.49 g/g–very close to the maximum theoretical values) [26,42,43,52], providing evidence that trials of *Saccharomyces cerevisiae* under full aerobic conditions can also lead to the significant production of ethanol. In Table 3, the metrics for ethanol production and the initial sugar content of various *Saccharomyces cerevisiae* strains cultured on different carbon sources are presented and compared with results from the current study.

**Table 3.** Metrics for ethanol production of *Saccharomyces cerevisiae* different strains cultured on various carbon sources in comparison with results from the current study.

| Yeast Strain | Carbon Source | Initial Sugar (g/L) | EtOH (g/L) | Reference |
|---|---|---|---|---|
| *S. cerevisiae* AXAZ-1 | Microalgae biomass and raisin extract | 257 | 111 | [14] |
| *S. cerevisiae* AXAZ-1 | Pomegranate residue hydrolysate | 37 | 13 | [17] |
| *S. cerevisiae* LMBF-Y 16 | Grape must | 250 | 112 | [26] |
| *S. cerevisiae* LMBF-Y 18 | Grape must | 250 | 125 | [26] |
| *S. cerevisiae* MAK-1 | Grape musts | 250 | 106–119 | [52] |
| Bakers' yeast | Carob pod | 200–350 | 62 | [54] |
| *S. cerevisiae* | Carob Pod Extracts | 200 | 95 | [55] |
| *S. cerevisiae* NP01 | Sweet sorghum juice | 280–300 | 134 | [56] |
| *S. cerevisiae* BY4741 | Sweet sorghum juice | 278.6 | 113 | [57] |
| *S. cerevisiae* NP01 | Sucrose | 280 | 95 | [58] |
| *S. cerevisiae* 27817 | Glucose | 50–200 | 5–91 | [59] |
| *S. cerevisiae* 2.399 | Glucose | 32 | 13 | [60] |
| *S. cerevisiae* 24860 | Glucose | 150 | 48 | [61] |
| *S. cerevisiae* CMI237 | Sugar | 160 | 70 | [62] |
| *S. cerevisiae* EC1118 | Grape must | 280 | 105 | [63] |
| *S. cerevisiae* DBVPG 1014 | Grape must | 270 | 115 | [64] |
| *S. cerevisiae* BP2-17 | Grape must | 225 | 89 | [21] |
| *S. cerevisiae* BP2-33 | Grape must | 225 | 89 | [21] |
| *S. cerevisiae* PP2-22 | Grape must | 225 | 86 | [21] |
| *S. cerevisiae* Mpr2-42 | Grape must | 225 | 87 | [21] |
| *S. cerevisiae* PR50 | Grape must | 220 | 72 | [24] |
| *S. cerevisiae* PR543 | Grape must | 220 | 85 | [24] |

**Table 3.** *Cont.*

| Yeast Strain | Carbon Source | Initial Sugar (g/L) | EtOH (g/L) | Reference |
|---|---|---|---|---|
| *S. cerevisiae* LMBF Y-54 | Glucose | 200 | 68.0 | Current study |
| *S. cerevisiae* Cross X | Glucose | 200 | 82.2 | Current study |
| *S. cerevisiae* Passion Fruit | Glucose | 200 | 91.1 | Current study |
| *S. cerevisiae* LMBF Y-54 | Grape must Assyrtiko | 222 | 106.3 | Current study |
| *S. cerevisiae* LMBF Y-54 | Grape must Mavrotragano | 214 | 99.7 | Current study |
| *S. cerevisiae* Passion Fruit | Grape must Assyrtiko | 222 | 102.7 | Current study |
| *S. cerevisiae* Passion Fruit | Grape must Mavrotragano | 214 | 97.8 | Current study |

### 3.3. Volatile Compounds–Wine Sensory Evaluation

At the end of grape must fermentations with the commercial Passion Fruit strain and the wild-type LMBF Y-54 strain cultured on Assyrtiko and Mavrotragano grape musts, the volatile compounds were analyzed with GC–MS (SPME) analysis. More than 100 volatiles have been identified, belonging to five major chemical group compounds from acids, aldehydes, ketones, esters and higher alcohols and as far as Assyrtiko is concerned the results are in agreement with other studies of the literature [7]. Mavrotragano is a rare indigenous *Vitis vinifera* Greek grape variety mostly unexploited, while the literature regarding its volatile composition is limited. The average (%) contribution of these groups of compounds for both Assyrtiko and Mavrotragano wines are presented in Figure 7. The ester group was comprised by the highest number of volatile compounds [65] and had the higher percentage of contribution to the volatile content of wines followed by higher alcohols. Esters are considered a significant group of volatile compounds as they pose a major impact on flavor and aroma of alcoholic beverages, while a plethora of esters is also produced during fermentation as a result of yeast metabolism [66,67]. Esters mainly contribute to the aromatic profile of wine via fruity and floral odors [65,67]. In the present study, no significant differences were found among the relative contents of esters between the commercial (Passion Fruit) and wild-type (LMBF Y-54) yeast strain in both white and red wines.

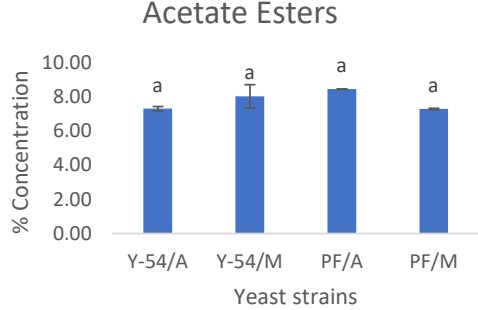

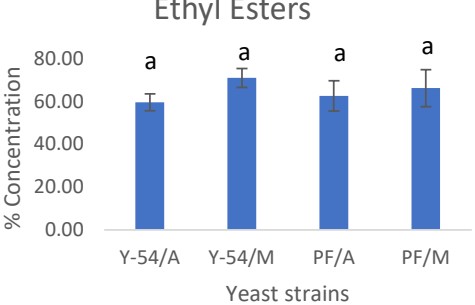

**Figure 7.** *Cont.*

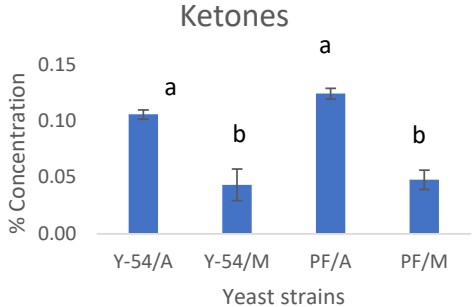

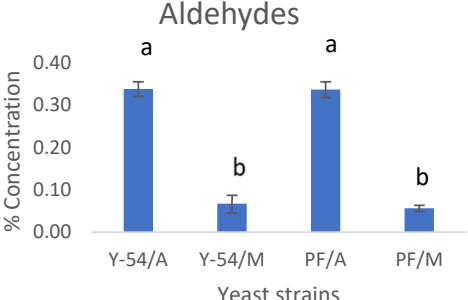

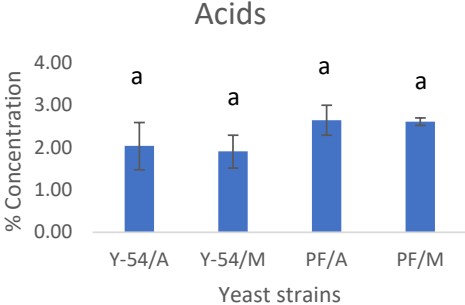

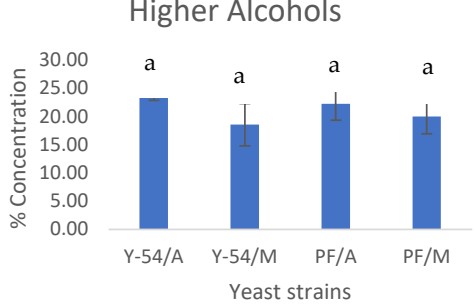

**Figure 7.** *Cont.*

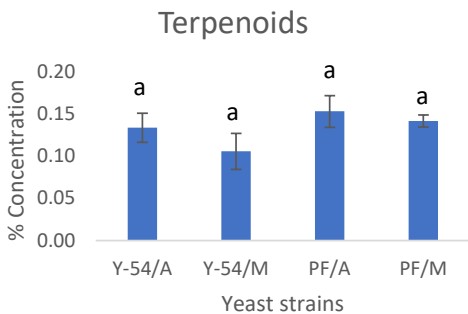

**Figure 7.** Percent concentrations of the most important groups of volatile compounds: esters (ethyl, acetic), ketones, aldehydes, fatty acids, higher alcohols and terpenoids. Y-54/A: Assyrtiko wine made with *S. cerevisiae* LMBF Y-54; Y-54/M: Mavrotragano wine made with *S. cerevisiae* LMBF Y-54; PF/A: Assyrtiko wine made with *S. cerevisiae* Passion Fruit; PF/M: Mavrotragano wine made with *S. cerevisiae* Passion Fruit. Columns with different letters are significantly different ($p < 0.05$).

Statistical differences were only observed among the white (Assyrtiko) and red wines (Mavrotragano) in terms of their ketone and aldehyde contents. In more detail, significantly lower contribution of the two groups of compounds mentioned above were observed in red wines irrespectively of the yeast strain used. This can be attributed to the higher phenolic content of red wines and consequently their higher antioxidant activity which exerts a protective role on oxidation phenomena, thus preventing the oxidation of carbonyl compounds to their corresponding acids [7]. Interestingly, the volatile composition of either red or white wines produced by the LMBF Y-54 and commercial strains did not differ significantly, verifying the high potential of the novel Y-54 strain to produce high-quality wines. The results from the GC–MS analysis are in line with the sensory data. Specifically, no statistically significant differences were observed between the results for the "control" wines (*viz.* the ones produced with the commercial Passion Fruit yeast strain implicated in the fermentation) and the wines made with the newly isolated and studied yeast strain (LMBF Y-54). Statistically significant differences were only observed concerning fresh and dried fruit odors; however, these differences were between red and white wines and not between the yeast stains studied in this experiment. Moreover, the aroma of all wines examined was of medium intensity with a fresh fruity and floral character and absence of defects, while the overall quality was rated as very high confirming the high potentiality of the novel yeast strain on the wine-making process (Figure 8).

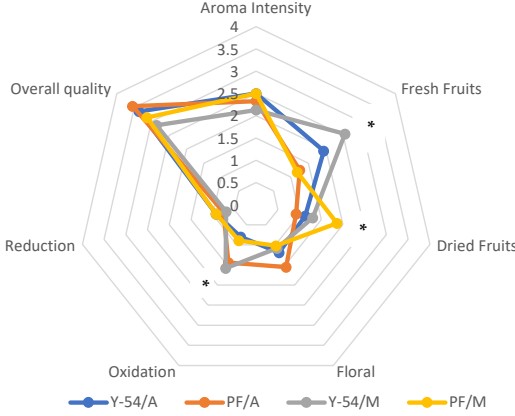

**Figure 8.** Sensory data of wine samples. Y-54/A: Assyrtiko wine made with *S. cerevisiae* LMBF Y-54; Y-54/M: Mavrotragano wine made with *S. cerevisiae* LMBF Y-54; PF/A: Assyrtiko wine made with *S. cerevisiae* Passion Fruit; PF/M: Mavrotragano wine made with *S. cerevisiae* Passion Fruit. *: Significant differences were found only between the values of red and white wine samples.

## 4. Conclusions

The exploitation of novel yeast strains in wine-making is considered as an important path in order to perform targeted must fermentation for unique and aromatic products. On the other hand, isolation of yeast strains capable of converting sugars into ethanol with high conversion yields (near to the maximum theoretical one that is =0.51 g of ethanol per g of consumed sugar) and high product concentrations (i.e., $\geq$100 g/L) presents significant importance for bioethanol-producing facilities. Targeted yeast metabolic compounds such as ethanol and glycerol are of paramount importance in wine-making as they provide the necessary alcohol content along with a thick mouthfeel in wine products.

The current study verifies the successful application of the novel and not previously systematically studied yeast strain *Saccharomyces cerevisiae* LMBF Y-54, for both the production of bioethanol from glucose-based media and red and white must fermentation. Interestingly, the cultivation under aerobic conditions resulted in lower production (in terms of both absolute (g/L) and relative (g of ethanol produced per g of sugar consumed)) compared to the trial performed under microaerophilic/anaerobic environment. This could be a first indication that this specific strain could be a good candidate for the production of wines with lower ethanol content, as suggested in other literature reports [24], producing wines of good quality while maintaining the "low-alcohol" phenotype. Likewise, in both tested musts from Santorini Island, the mentioned strain (LMBF Y-54) succeeded in dominating the indigenous yeast strains during all the wine-making fermentation processes. The novel strain compared to the commercial one showed optimum technological characteristics verifying its suitability for wine production and thus posing great potential for industrial applications. Nevertheless, further studies based on genomic approaches can provide supplementary information regarding the fermentative potential of the yeast strain and thus further valorize the obtained knowledge in future wine-making bioprocesses.

**Author Contributions:** Conceptualization: S.P.; Methodology, K.B., M.D., A.T. and S.K.; Validation: S.K., G.-J.E.N. and S.P.; Formal Analysis: M.D., A.T. and S.K.; Investigation: K.B. and M.D.; Resources: S.P. and G.-J.E.N.; Writing—Original Draft Preparation: M.D., A.T. and S.P.; Writing—Review & Editing: M.D., A.T., S.K. and S.P.; Supervision: S.K. and S.P.; Project Administration: G.-J.E.N., S.K. and S.P.; Funding acquisition: G.-J.E.N. and S.P. All authors have read and agreed to the published version of the manuscript.

**Funding:** The current investigation was financially supported by the project entitled "Exploitation of new natural microbial flora from Greek origin amenable for the production of high-quality wines" (Acronym: Oenovation, project code T1EΔK-04747) financed by the Ministry of National Education and Religious Affairs, Greece (project action: "Investigate–Create–Innovate 2014–2020, Intervention II").

**Institutional Review Board Statement:** Not applicable.

**Informed Consent Statement:** Not applicable.

**Conflicts of Interest:** The authors declare no conflict of interest.

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
