# Peer review of "Trials of Commercial- and Wild-Type Saccharomyces cerevisiae Strains under Aerobic and Microaerophilic/Anaerobic Conditions: Ethanol Production and Must Fermentation from Grapes of Santorini (Greece) Native Varieties"

_fermentation, doi:10.3390/fermentation8060249_

Round 1

Reviewer 1 Report

The manuscript under evaluation assesses the biochemical and technological properties of 3 commercial and 4 wild-type yeasts concerning the comparison of strains’ performance and productivity. The authors conducted several studies and concluded that the Saccharomyces cerevisiae LMBF Y-54 (selected new wild yeast strain) could be applied successfully for both the production of bioethanol from glucose-based media and the red and White must fermentation and it exhibited great potential for future industrial applications. The discussion is fluent and extensive. However, the following minor revisions are required to improve the manuscript.

  • P3, L121: Please clarify the initial yeast population in the text. Did 5 ml taken from culture of 106cfu/ml. ( total initial inoculum 5 x 106cfu/ml), or there is 106cfu/ml in total of 5 ml inoculation (total initial inoculum 106cfu/ml) ?
  • P3, L142 – P4, L149: (under Analytical methods) Validation parameters for HPLC analysis (at least; app. range, LOD, LOQ, recovery and r2values) can be given. Or a published reference containing validation parameters.
  • P4, L176: Please explain the selection criteria (20% of colonies) of the colonies picked up for identification purposes. (size, colour, margin/edge, form, etc.).
  • P4, L177: If any extraction kit is used for total DNA extraction, please indicate it in this sentence.
  • P14, L397-398 glycerol results can be given in the Table or as supplementary data.
  • Both sentences in P15, L438-441, and P16, L:447-449 are synonyms. Please edit one of them or combine them in one sentence to avoid repetition.
  • P16, L447-454: more detail can be given about the sensory evaluation. How could you explain the result indicating that there are no significant statistical differences between red and white one and also between wild strain and control strain? According to panelists, whether novel yeast strain provides higher overall quality in red wine or white wine?

Author Response

Important note: All of our additions are highlighted with yellow fond.

Reviewer 1.

P4, L176: Please explain the selection criteria (20% of colonies) of the colonies picked up for identification purposes. (size, colour, margin/edge, form, etc.).

Thank you for your comment. Yes, from each sample, a number of colonies, randomly selected according to the representative sampling scheme of Harrigan and McCance. This sampling scheme is still being applied over time (Doulgeraki et al., 2010, Syrokou et al., 2020). The text has been enriched.

  • Harrigan, W.F.; McCance, M.E. Laboratory Methods in Food and Dairy Microbiology; Academic Press: London, UK, 1976; pp. 47–49
  • Doulgeraki, A.I.; Paramithiotis, S.; Kagkli, D.M.; Nychas, G.-J.E. Lactic acid bacteria population dynamics during minced beef storage under aerobic or modified atmosphere packaging conditions. Food Microbiol. 2010, 27, 1028–1034.
  • Syrokou MK, Themeli C, Paramithiotis S, Mataragas M, Bosnea L, Argyri AA, Chorianopoulos NG, Skandamis PN, Drosinos EH. Microbial Ecology of Greek Wheat Sourdoughs, Identified by a Culture-Dependent and a Culture-Independent Approach. Foods. 2020 Nov 4;9(11):1603. doi: 10.3390/foods9111603. PMID: 33158141; PMCID: PMC7694216.

P4, L177: If any extraction kit is used for total DNA extraction, please indicate it in this sentence.

Thank you for your comment. No we did not use any extraction kit. We have followed exactly the same protocol as our referred work (Bonatsou et al., 2018), which is based on the initial protocol of Querol et al. (1992). The text has been enriched.

  • Querol, A., Barrio, E., Huerta, T., and Ramon, D. (1992). Molecular monitoring of wine fermentations conducted by active dry yeast srtains. Appl. Environ. Microbiol. 58, 2948–2953.

P16, L447-454: more detail can be given about the sensory evaluation. How could you explain the result indicating that there are no significant statistical differences between red and white one and also between wild strain and control strain? According to panelists, whether novel yeast strain provides higher overall quality in red wine or white wine.

What we have reported is that significant differences were obtained between red and white wines but not between wild and commercial stains. The differences between white and red wines, as far as their sensory character is concerned, are not surprising since these two types of wines differ in their chemical composition and the winemaking process. The panelists rated the overall quality of the white wine produced by the novel yeast strain higher compared with the respective red wine. Figure 8 was added to the text to demonstrate the results.

Both sentences in P15, L438-441, and P16, L:447-449 are synonyms. Please edit one of them or combine them in one sentence to avoid repetition.

Both sentences were modified to avoid repetition

P3, L121: Please clarify the initial yeast population in the text. Did 5 ml taken from culture of 106cfu/ml. (total initial inoculum 5 x 106cfu/ml), or there is 106cfu/ml in total of 5 ml inoculation (total initial inoculum 106cfu/ml) ?

In each mL of pre-culture we have 106 cells, thus the inoculation per bottle was 5 x 106 cells,

P3, L142 – P4, L149: (under Analytical methods) Validation parameters for HPLC analysis (at least; app. range, LOD, LOQ, recovery and r2 values) can be given. Or a published reference containing validation parameters.

We do not consider that the elements requested for validation parameters of HPLC analysis are needed. We could potentially add a calibration curve of one of our compounds, but, indeed, I do not consider that it is needed.

  1. P14, L397-398 glycerol results can be given in the Table or as supplementary data.

Glycerol values have been added (see Tables 1 and 2).

Both sentences in P15, L438-441, and P16, L:447-449 are synonyms. Please edit one of them or combine them in one sentence to avoid repetition.

Corrections were made.

Reviewer 2 Report

Manuscript 1693747
Journal Fermentation
Title Trials of commercial- and wild-type Saccharomyces cerevisiae strains under aerobic and microaerophilic/anaerobic conditions: ethanol production and must fermentation from grapes of Santorini (Greece) native varieties
The manuscript entitled “Trials of commercial- and wild-type Saccharomyces cerevisiae strains under aerobic and microaerophilic/anaerobic conditions: ethanol production and must fermentation from grapes of Santorini (Greece) native varieties” describes the selection of wild yeasts for the must fermentation of Santorini native grape varieties. The topic is not novel and this kind of study have been largely published over time. Several aspects should be clarified. Discussion of the results is limited. For these reasons, a major revision is strongly suggested. Please follow the comments in the file.

Author Response

Important note: All of our additions are highlighted with yellow fond.

Reviewer 2

L176-177 Why 20% colonies were picked up from dilutions? Please add a reference for this choice

Thank you for your comment. Yes, from each sample, a number of colonies, randomly selected according to the representative sampling scheme of Harrigan and McCance. This sampling scheme is still being applied over time (see Doulgeraki et al., 2010 and Syrokou et al., 2020). The text has been enriched. Recent literature (Doulgeraki et al., 2010 and Syrokou et al., 2020) was added.

  • Harrigan, W.F.; McCance, M.E. Laboratory Methods in Food and Dairy Microbiology; Academic Press: London, UK, 1976; pp. 47–49
  • Doulgeraki, A.I.; Paramithiotis, S.; Kagkli, D.M.; Nychas, G.-J.E. Lactic acid bacteria population dynamics during minced beef storage under aerobic or modified atmosphere packaging conditions. Food Microbiol. 2010, 27, 1028–1034.
  • Syrokou MK, Themeli C, Paramithiotis S, Mataragas M, Bosnea L, Argyri AA, Chorianopoulos NG, Skandamis PN, Drosinos EH. Microbial Ecology of Greek Wheat Sourdoughs, Identified by a Culture Dependent and a Culture-Independent Approach. Foods. 2020 Nov 4;9(11):1603.

L177 Please briefly describe the DNA isolation protocol in the text

Thank you for your comment. We have followed exactly the same protocol as the cited work (Bonatsou et al., 2018), which is based on the initial protocol of Querol et al. (1992). The text has been enriched, and was supported by the more recent literature source (Bonatsou et al., 2018).

L180-188 Molecular biotyping was performed using a rep-PCR. Why authors used this method? Interdelta polymorphisms are largely used for this purpose. Please explain

Thank you for your comment. There is no doubt that the method of interdelta polymorphism is largely used to differentiate S. cerevisiae at strain level. Nevertheless, this method has been rarely applied for other non-Saccharomyces species which are dominant in the first steps of winemaking, for this reason we have applied a rep-PCR method as we were able to unambiguously discriminate between genotypes of different species and go up to strain level.

L187-188 Please add more details related to the cluster analysis. Please add the similarity threshold used to discriminate different strains, the marker used, the stability of the molecular profiles and so on.

Thank you for you comment. More details have been added (line 238-241).

L161 Please note that in HS-SPME-GC-MS analysis SPME fibers are used. Please add more details on the incubation of the fiber, the desorption of volatile compounds and GC-MS analysis with a particular focus on mass spectra analysis. Thanks.

Requested elements are given in significant details (lines 186-203).

L166 semi-quantitative…Why authors did not consider a quantitative analysis using known standards? Quantitative analysis is important for this kind of study because volatile compounds are associated with sensory properties of wines. Selected strains can produce different VOCs, giving peculiar sensory traits to the wines. Please explain

Our aim was to compare the volatile profiles of the wines produced by the novel yeasts with those produced by the commercial ones and not to measure the absolute amounts of each volatile compound. For comparison purposes, this semi-quantitative method which is based on % area contribution of each compound was considered appropriate.

L190-191 Twelve panelist? It is a limited number for this kind of study. Please explain

Twelve trained panelists are an adequate number of judges for this kind of studies. They performed three replicate tastings for each sample. Similar results we have obtained with 12 (to 15) panelists which are published in well-respected peer reviewed journals.

  1. Kallithraka, S., Bakker, J. and Clifford, M.N. (1997) Evaluation of bitterness and astringency of (+)-catechin and (-)-epicatechin in red wine and in model solution. Journal of Sensory Studies, 12, 25-37.
  2. Kallithraka, S., Bakker, J. and Clifford, M.N. (1997) Red wine and model wine astringency as affected by malic and lactic acid. Journal of Food Science, 62(2), 416-420.
  3. Kallithraka, S., Bakker, J. and Clifford, M.N. (1997) Effect of pH on astringency in model solutions and wines. Journal of Agricultural and Food Chemistry, 45(6), 2211-1116.
  4. Kallithraka, S., Kim, D., Tsakiris, A., Paraskevopoulos, I., Soleas, G. (2011), Sensory assessment and chemical measurement of astringency of Greek wines: Correlations with analytical polyphenolic composition, Food Chemistry 126 (4) pp. 1953-1958
  5. Kallithraka, S., Kotseridis, Y., Kyraleou, M., Proxenia, N., Tsakiris, A., Karapetrou, G. (2015). Analytical phenolic composition and sensory assessment of selected rare Greek cultivars after extended bottle ageing. Journal of the Science of Food and Agriculture, 95, 1638-1647.
  6. Kyraleou, M., Kallithraka, S., Chira, K., Tzanakouli, E., Ligas, I., Kotseridis Y. (2015) Differentiation of Wines Treated with Wood Chips Based on Their Phenolic Content, Volatile Composition, and Sensory Parameters, Journal of Food Science 2015, Vol. 80 (12), 2701-2710.
  7. Kyraleou, M., Kotseridis, Y., Koundouras, S., Chira, K., Teissedre PL, Kallithraka, S. (2016) Effect of irrigation regime on perceived astringency and proanthocyanidin composition of skins and seeds of Vitis vinifera L. cv. Syrah grapes under semiarid conditions Food Chemistry 203 292–300.

L197 Please add more details on the 5-point scale.

The two ends of the scale were marked with zero (left) and maximum (right) intensities of the attributes. Added at the text.

L429-441 Please add statistical analysis section, statistical analysis in Figure 7 (add different letters or symbols to indicate statistical differences) and P value

Figure 7 was changed to include statistics

L447-454 Please add more details on the results of the sensory analysis. Authors used a 5-point scale but no mention is reported in the results.

Figure 8 was added in the manuscript to clarify and present the results

425-426 Are there differences in the profile of VOCs between two cultivars? Is it possible a quantitative comparison?

Νο statistical differences were obtained between the volatile composition of neither white nor red wines produced by the LMBF Y-54 and commercial strains. Mentioned in the text.

L467-469 Is it possible to use this strain for the production of low-alcohol wines? Please add a perspective.

This could be an indication that this specific strain could be a good candidate for the production of wines with lower ethanol content (comment added at the text)

Which is the novelty of the work? Similar papers have been published (see doi.org/10.1016/j.ijfoodmicro.2010.09.009; doi.org/10.3390/fermentation6020043; doi.org/10.1007/s11274-011-0744-0; doi.org/10.3390/fermentation5040087). Please better describe the novelty of the work in the Introduction section and in the abstract.

Additions and comments according to the referee’s suggestions as regards the novelty of the work were added in “Abstract” (lines 11-15) and “Introduction” sections (lines 81-93 and 99-102).

The selection of wild strains is not clear. Why authors selected S. cerevisiae LMBF Y54 and Passion Fruit for the high ethanol yield under aerobic condition and then performed trials under microaerobic/anaerobic conditions? It seems not reasonable. Please explain this choice and add a justification in the manuscript.

Justifications on the choice of the yeasts in order to perform the microaerophilic/anaerobic trials are given (lines (452-466).

L2-5 Title is too long and could be shortened. Please revise and follow MDPI style.

We consider that the title represents pur work, thus we would like to maintain it for our m/s.

L11-31 Rewrite. Please add more details about the novelty of the work, the selection of wild strains, technological properties of strains with a focus on aromatic profile and fermentation kinetics.

The abstract has been revised as: Subsequently, the “novel” strain that presented the best technological characteristics as regards its sugar consumption and alcohol production properties (viz. LMBF Y54) and the commercial strain that equally presented the best previously mentioned technological characteristics (viz. Passion Fruit) were further selected for wine-making process. (lines 26-29)

L42-43 Rewrite. It is not correct in English.

The sentence has been rephrased.

L57-58 Please explain the term high gravity fermentation in the introduction. Very high gravity fermentation is “the preparation and fermentation to completion of mashes containing 27 g or more dissolved solids per 100 g mash” (Thomas et al. 1993).

Thomas KC, Hynes SH, Jones AM, Ingledew WM (1993) Production of fuel alcohol from wheat by VHG technology: effect of sugar concentration and fermentation temperature. Appl Biochem Biotechnol 43:211–226

The manuscript has been revised and the suggested reference has been included (see lines 65-71)/

Regarding bioprocess optimization, research mainly focuses on the technological op-timization of bioprocesses, like i.e. the effect of the agitation/aeration upon the process, the “very high-gravity” application of the fermentation (viz. “the preparation and fer-mentation to completion of mashes containing 27 g or more dissolved solids per 100 g mash”), the utilization of “new” fermentation feedstocks (e.g. algae, crude glycerol, food-processing wastes, hemicellulose hydrolysates), and the development of innova-tive bioreactor designs.

L70-71 Please introduce papers dealing with the selection of wild S. cerevisiae strains for high gravity fermentation, robustness, technological properties and high-quality wines. Add relevant references.

Selected S. cerevisiae strains of for high gravity fermentation, robustness and unique technological properties compared with commercial strains and strains of the current study can also be found in Table 3.

The following references in regard to high gravity fermentation, robustness, technological properties and high-quality wines are included in the introduction section:

  1. Sarris, D.; Papanikolaou, S. Biotechnological production of ethanol: Biochemistry, processes and technologies. Engineering in Life Sciences 2016, 16, 307-329, doi:https://doi.org/10.1002/elsc.201400199.
  2. Roukas, T.; Kotzekidou, P. From food industry wastes to second generation bioethanol: a review. Reviews in Environmental Science and Bio/Technology 2022, 21, 299-329, doi:10.1007/s11157-021-09606-9.
  3. Tsolcha, O.N.; Patrinou, V.; Economou, C.N.; Dourou, M.; Aggelis, G.; Tekerlekopoulou, A.G. Utilization of Biomass Derived from Cyanobacteria-Based Agro-Industrial Wastewater Treatment and Raisin Residue Extract for Bioethanol Production. Water 2021, 13, 486.
  4. Thomas, K.C.; Hynes, S.H.; Jones, A.M.; Ingledew, W.M. Production of fuel alcohol from wheat by VHG technology. Applied Biochemistry and Biotechnology 1993, 43, 211-226, doi:10.1007/BF02916454.
  5. Mavrommati, M.; Daskalaki, A.; Papanikolaou, S.; Aggelis, G. Adaptive laboratory evolution principles and applications in industrial biotechnology. Biotechnology Advances 2022, 54, 107795, doi:https://doi.org/10.1016/j.biotechadv.2021.107795.
  6. Dourou, M.; Economou, C.N.; Aggeli, L.; Janák, M.; Valdés, G.; Elezi, N.; Kakavas, D.; Papageorgiou, T.; Lianou, A.; Vayenas, D.V.; et al. Bioconversion of pomegranate residues into biofuels and bioactive lipids. Journal of Cleaner Production 2021, 323, 129193, doi:https://doi.org/10.1016/j.jclepro.2021.129193.
  7. Camarasa, C.; Chiron, H.; Daboussi, F.; Della Valle, G.; Dumas, C.; Farines, V.; Floury, J.; Gagnaire, V.; Gorret, N.; Leonil, J.; et al. INRA's research in industrial biotechnology: For food, chemicals, materials and fuels. Innovative Food Science and Emerging Technologies 2018, 46, 140-152, doi:10.1016/j.ifset.2017.11.008.
  8. Mohd Azhar, S.H.; Abdulla, R.; Jambo, S.A.; Marbawi, H.; Gansau, J.A.; Mohd Faik, A.A.; Rodrigues, K.F. Yeasts in sustainable bioethanol production: A review. Biochemistry and Biophysics Reports 2017, 10, 52-61, doi:https://doi.org/10.1016/j.bbrep.2017.03.003.

L104-105 Here and throughout the manuscript replace c. with ca.

Done.

L107-108 How many flasks were used? Three replicates?

The response was found already in the initial version, and now is found in lines 253-257.  

L117-118 How microaerobic/anaerobic condition was assessed?

We followed the experimental procedure found in literature (see ref. 14). See also lines 135 and 467-470.

L148-149 Please add the range of concentration used to build the calibration curve for each compound.

We do not consider that the elements requested for validation parameters of HPLC analysis are needed. We could potentially add a calibration curve of one of our compounds, but, indeed, I do not consider that it is needed.

L157 Replace the cultures with the incubation.

We disagree with the comment. We perform cultures.

L187-188 Please add more details related to the cluster analysis. Please add the similarity threshold used to discriminate different strains, the marker used, the stability of the molecular profiles and so on.

Done; see lines 238-241.

L198-202 Is it possible to include statistical analysis in this kind of paper? Statistical analysis is lacking.

Please see lines 253-257. All of our kinetics and linear regressions (see Figs 1-6) demonstrate the repeatability of our kinetics.

Figure 1-6 Please move these figures in the Results and Discussion section. Thanks.

You have right. Nevertheless, we have not performed the paper lay out.

L263-270 Delete. Please explain abbreviations at the first mention in the manuscript or in the Table/Figure caption.

We consider that the abbreviations and units are needed and should be maintained in the revised m/s.  

L272-273 Rewrite. Replace with a new title. It is not clear.

Done; see lines 327-328.

L280 Please better describe the Crabtree effect. It is important. Please add the critical value and the metabolic pathway involved with relevant references.

Done; see lines 335-351.

L301-302 Please better describe in the text the phenomenon “ethanol make-accumulate-consume” and the link with the X parameter.

Done; see lines 372-383.

L308-309 Please add a discussion of these results in comparison to literature. In particular ethanol yields of Crabtree positive strains should be reported in %. The papers doi.org/10.1016/j.fm.2021.103893 and doi.org/10.1016/j.ijfoodmicro.2019.02.022 could be included for your analysis.

Discussions were done (see lines 389-396). The ref. doi.org/10.1016/j.ijfoodmicro.2019.02.022 was not added since it was irrelevant with the discussion.

L323-324 Please discuss the very high gravity fermentation. During VHG fermentation, yeasts are subjected to different stresses (thermal stress, oxidative stress and osmotic stress). The addition of N-acetyl-L-cysteine (NAC) (doi.org/10.1038/s41598-018-31558-4) or Mg2+ and peptone (doi.org/10.1007/s10529-006-9220-6) can be added to improve stress tolerance in S. cerevisiae. Please discuss these aspects.

Discussions were done (see lines 421-431).

L331-332 This result could be due to different environmental stresses. Please discuss.

Supplementary discussions are done (see lines 445-450).

L341-346 Please explain these discrepancies. Are they related to tested strains? Are they related to experimental conditions?

Text was re-phrased (see lines 438-444).

L385-386 Please add data on the ratio Ethanol/glycerol (purity of the fermentation) for each strain.

Elements concerning glycerol production were added (see Tables 1 and 2).

L411-412 Please discuss the results of the papers included at L412. Discussion is limited.

Discussions are added (see lines 532-547).

Table 3 Please revise table 3. Other S. cerevisiae strains fermenting grape must should be included with relevant references. Delete other matrices and the associated references

We disagree with the point of view of the referee, and thus we have maintained the results dealing with ethanol production from wastes/residues. On the other hand we have added references requested (ethanol production from musts).

425-426 Are there differences in the profile of VOCs between two cultivars? Is it possible a quantitative comparison?

Done.

L429-441 Please add statistical analysis section, statistical analysis in Figure 7 (add different letters or symbols to indicate statistical differences) and P value.

Done.

  1. L447-454 Please add more details on the results of the sensory analysis. Authors used a 5-point scale but no mention is reported in the results.

Done.

L467-469 Is it possible to use this strain for the production of low-alcohol wines? Please add a perspective.

Done.

Round 2

Reviewer 2 Report

Authors revised the original manuscript according to previous comments. However, several comments are still partially addressed. Please follow the comments in the file.

Author Response

We would like to thank the reviewer for their useful comments that helped us improve our manuscript. Below you will find the reply to each reviewer comment.

Authors revised the original manuscript according to previous comments. However, several comments are still partially addressed. Please follow the comments below:

  1. L190-191 Twelve panelist? It is a limited number for this kind of study. Please explain

Twelve trained panelists are an adequate number of judges for this kind of studies. They performed three replicate tastings for each sample. Similar results we have obtained with 12 (to 15) panelists which are published in well-respected peer reviewed journals.

  • Kallithraka, S., Bakker, J. and Clifford, M.N. (1997) Evaluation of bitterness and astringency of (+)-catechin and (-)-epicatechin in red wine and in model solution. Journal of Sensory Studies, 12, 25-37.
  • Kallithraka, S., Bakker, J. and Clifford, M.N. (1997) Red wine and model wine astringency as affected by malic and lactic acid. Journal of Food Science, 62(2), 416-420.
  • Kallithraka, S., Bakker, J. and Clifford, M.N. (1997) Effect of pH on astringency in model solutions and wines. Journal of Agricultural and Food Chemistry, 45(6), 2211-1116.
  • Kallithraka, S., Kim, D., Tsakiris, A., Paraskevopoulos, I., Soleas, G. (2011), Sensory assessment and chemical measurement of astringency of Greek wines: Correlations with analytical polyphenolic composition, Food Chemistry 126 (4) pp. 1953-1958
  • Kallithraka, S., Kotseridis, Y., Kyraleou, M., Proxenia, N., Tsakiris, A., Karapetrou, G. (2015). Analytical phenolic composition and sensory assessment of selected rare Greek cultivars after extended bottle ageing. Journal of the Science of Food and Agriculture, 95, 1638-1647.
  • Kyraleou, M., Kallithraka, S., Chira, K., Tzanakouli, E., Ligas, I., Kotseridis Y. (2015) Differentiation of Wines Treated with Wood Chips Based on Their Phenolic Content, Volatile Composition, and Sensory Parameters, Journal of Food Science 2015, Vol. 80 (12), 2701-2710.
  • Kyraleou, M., Kotseridis, Y., Koundouras, S., Chira, K., Teissedre PL, Kallithraka, S. (2016) Effect of irrigation regime on perceived astringency and proanthocyanidin composition of skins and seeds of Vitis vinifera L. cv. Syrah grapes under semiarid conditions Food Chemistry 203 292–300.

L244-245 Please include two of there references in order to support your choice.

Reply: The following references have been included in the manuscript.

  • Kyraleou, M., Kallithraka, S., Chira, K., Tzanakouli, E., Ligas, I., Kotseridis Y. (2015) Differentiation of Wines Treated with Wood Chips Based on Their Phenolic Content, Volatile Composition, and Sensory Parameters, Journal of Food Science 2015, Vol. 80 (12), 2701-2710.
  • Kyraleou, M., Kotseridis, Y., Koundouras, S., Chira, K., Teissedre PL, Kallithraka, S. (2016) Effect of irrigation regime on perceived astringency and proanthocyanidin composition of skins and seeds of Vitis vinifera L. cv. Syrah grapes under semiarid conditions Food Chemistry 203 292–300.

  1. 425-426 Are there differences in the profile of VOCs between two cultivars? Is it possible a quantitative comparison?

Νο statistical differences were obtained between the volatile composition of neither white nor red wines produced by the LMBF Y-54 and commercial strains. Mentioned in the text.

Please indicate the lines of this part. Thanks.

Reply: Lines 615-618 and 622-624.

  1. L70-71 Please introduce papers dealing with the selection of wild S. cerevisiae strains for high gravity fermentation, robustness, technological properties and high-quality wines. Add relevant references.

Selected S. cerevisiae strains of for high gravity fermentation, robustness and unique technological properties compared with commercial strains and strains of the current study can also be found in Table 3.

Please introduce and discuss reference 22 and 42 in the Introduction section (lines 81-93)

Reply: Reference No 22 was included in the introduction section while it has also been included in line 84.

Ref. No 42 has been introduced and discussed in the introduction section (Lines 88-90).

  1. L148-149 Please add the range of concentration used to build the calibration curve for each compound.

We do not consider that the elements requested for validation parameters of HPLC analysis are needed. We could potentially add a calibration curve of one of our compounds, but, indeed, I do not consider that it is needed.

L173-175 Please add the range of concentration used to build the calibration curve for each compound. It is important

Reply: See lines 178-180. The range of concentration of the compounds was between 0.0 and 20.0 g/L. Please see below a characteristic calibration curve for glucose. Similar calibration curves are also generated in order to quantify the other fermentation substrate (fructose) and products (ethanol and glycerol).

  1. 5. L157 Replace the cultures with the incubation.

We disagree with the comment. We perform cultures.

Replace “occurred throughout the cultures” with “were maintained”

Reply: The sentence has been revised accordingly please see Line 188.

  1. L198-202 Is it possible to include statistical analysis in this kind of paper? Statistical analysis is lacking.

Please see lines 253-257. All of our kinetics and linear regressions (see Figs 1-6) demonstrate the repeatability of our kinetics.

L253-257 Please add the statistical analysis applied to Figure 7. Thanks

Reply: Statistical analysis is described (lines 263-266)

  1. Figure 1-6 Please move these figures in the Results and Discussion section. Thanks.

You have right. Nevertheless, we have not performed the paper lay out.

Authors should revise the layout of the manuscript including Figure 1-6 in the Results and Discussion section. Please move figures in this section. Thanks

Reply: The figures have been included in the Results and Discussion section

  1. L301-302 Please better describe in the text the phenomenon “ethanol make-accumulate-consume” and the link with the X parameter.

Done; see lines 372-383.

L375-380 Please add a reference for this statement. Thanks

Reply: The appropriate references have been included [12,41]

  • Sarris, D.; Papanikolaou, S. Biotechnological production of ethanol: Biochemistry, processes and technologies. Engineering in Life Sciences 2016, 16, 307-329, doi:https://doi.org/10.1002/elsc.201400199.
  • Hagman, A.; Säll, T.; Compagno, C.; Piskur, J. Yeast "make-accumulate-consume" life strategy evolved as a multi-step process that predates the whole genome duplication. PLoS One 2013, 8, e68734, doi:10.1371/journal.pone.0068734.

  1. L308-309 Please add a discussion of these results in comparison to literature. In particular ethanol yields of Crabtree positive strains should be reported in %. The papers doi.org/10.1016/j.fm.2021.103893 and doi.org/10.1016/j.ijfoodmicro.2019.02.022 could be included for your analysis.

Discussions were done (see lines 389-396). The ref. doi.org/10.1016/j.ijfoodmicro.2019.02.022 was not added since it was irrelevant with the discussion.

Please enrich the discussion with ethanol yields from glucose based-media and musts with non-Saccharomyces yeasts (Crabtree positive yeasts) such as Starmerella bacillaris (synonym C. zemplinina). The papers doi.org/10.1016/j.ijfoodmicro.2019.02.022 and doi.org/10.1016/j.ijfoodmicro.2019.02.022 could be included for your analysis.

Reply: The suggested paper (doi.org/10.1016/j.ijfoodmicro.2019.02.022) was added. Please see additions in lines 345-349.

  1. L341-346 Please explain these discrepancies. Are they related to tested strains? Are they related to experimental conditions?

Text was re-phrased (see lines 438-444).

L438-444 Rewrite. It is not clear

Reply: The sentence has been revised.

  1. L385-386 Please add data on the ratio Ethanol/glycerol (purity of the fermentation) for each strain.

Elements concerning glycerol production were added (see Tables 1 and 2).

Please replace Glol with Glyc in Table 1, 2. Thanks

Reply: Tables 1 & 2 have been revised.

  1. Table 3 Please revise table 3. Other S. cerevisiae strains fermenting grape must should be included with relevant references. Delete other matrices and the associated references

We disagree with the point of view of the referee, and thus we have maintained the results dealing with ethanol production from wastes/residues. On the other hand, we have added references requested (ethanol production from musts).

Please delete other matrices and the associated references in Table 3. Thanks

Reply:  Other matrices and the associated references has been deleted from Table 3 as suggested.

  1. 13. L447-454 Please add more details on the results of the sensory analysis. Authors used a 5-point scale but no mention is reported in the results.

Done.

L599-602 Please better describe the results of the new Figure 8. Thanks

Reply:  Lines 615-617 and 622-624 were added

Round 3

Reviewer 2 Report

Authors addressed reviewer's comments. Minor revisions are suggested below:

1) L324-329 Rewrite. It is not clear 

2) L622-624 Rewrite. It is not clear.

3) Please revise English language throughout the manuscript

Author Response

1) L324-329 Rewrite. It is not clear.

Lines were not changed. We consider that they were clear.

2) L622-624 Rewrite. It is not clear.

Lines were not changed. We consider that they were clear.

3) Please revise English language throughout the manuscript

Minor changes were made in the revised mnuscript. Please see changed in yellow fond.